# Brief communication: An alternative method for estimating the scavenging efficiency of black carbon by meltwater over sea ice

Tingfeng Dou[1,2], Zhiheng Du[2], Shutong Li[2], Yulan Zhang[2], Qi Zhang[4], Mingju Hao[5], Chuanjin Li[2], Biao Tian[4], Minghu Ding[4], Cunde Xiao[3,2,4]

[1]College of Resources and Environment, University of Chinese Academy of Sciences, Beijing 100049, China
[2]State Key Laboratory of Cryospheric Sciences, Chinese Academy of Sciences, Lanzhou 730000, China
[3]State Key Laboratory of Land Surface Processes and Resource Ecology, Beijing Normal University, Beijing 100875, China
[4]Institute of Tibetan Plateau and Polar Meteorology, Chinese Academy of Meteorological Sciences, Beijing 100081, China
[5]Ministry of Education Key Laboratory for Earth System Modeling, Department of Earth System Science, Tsinghua
University, Beijing 100084, China

*Correspondence to*: Tingfeng Dou (doutf@ucas.ac.cn)

**Abstract. The meltwater scavenging coefficient (MSC) of black carbon (BC) is a ~~key~~ crucial parameter in snow and sea ice model, as it determines the BC enrichment in the surface layer of melting snow over sea ice and therefore modulates the BC-snow-albedo feedbacks. We present a new method for MSC estimation by sampling the melt-**
**refreeze ice layer~~ice layer~~ that is produced from refreezing of the meltwater within snowpack~~within the snowpack~~ and its overlying snow and measuring their physical characteristics ~~in the snowpits~~ in Elson Lagoon– northeast of Barrow, Alaska during the melting season. The bias of estimated MSC ranges from -5.4% to 7.3%, which is not exactly dependent on the degree of ablation. The average MSC value (14.5%±2.6%) calculated by this proposed method is slightly lower than that derived from the repeating sampling (RS) method in Elson Lagoon, while still is**
**within its best estimate range. Further estimation demonstrates that the MSC in Canada Basin (23.6%±2.1%) is close to that in Greenland (23.0%±12.5%~~,~~) and larger than that in Chukchi Sea (17.9%±5.0%) on the northwest of Barrow. Elson Lagoon has the lowest MSC (14.5%±2.6%) in the study areas. We~~It is~~ conclude~~d~~ that MSC exhibited a regional difference in the western Arctic during the sampling period. The method suggested in this study provides a possible approach for large-scale measurements of MSC over the sea-ice area in the Arctic~~.~~, of course, this method depends on**
**the presence of a melt-refreeze ice layer in the observation area.**

## 1 Introduction

BC is among the most efficient particulate species at absorbing visible light, which can reduce the surface albedo and potentially accelerate snow melting (Flanner et al., 2007; Goldenson et al., 2012; Dou et al., 2012; 2017). Previous studies
suggested an annual-mean radiative forcing of 0.1–0.3 W m$^{-2}$ over the Arctic region from BC deposition (Flanner et al., 2009; Jiao et al., 2014). However, significant~~large~~ uncertainties still exist in the sea ice region due to lack of field measurements and poor understanding of BC enrichment by overlying snow melting.

The enrichment of BC in melting snow largely depends on MSC, as it reflects the ratio of BC concentration in the meltwater
departing the snow layer to the bulk concentration in the exact layer (Flanner et al., 2007). MSC which leads to enhanced
concentrations of BC in surface snow ~~has~~ is ~~been found to be~~ considerably less than 100% by very few previous studies (e.g.,
Conway et al., 1996; Xu et al., 2012; Doherty et al., 2013). In present snow and sea ice models (e.g., Flanner et al., 2007;
Goldenson et al., 2012; Holland et al., 2012), MSC is valued as a constant of 20% and 3% for hydrophilic BC and
hydrophobic BC, respectively, which were derived from the observations conducted at Snowdome (2050 m) of the mid-
latitude Blue Glacier (Conway et al., 1996).

More recently, the MSC of BC was re-evaluated based on the field measurements in Elson Lagoon (Barrow, Alaska) and at
Dye-2 station (Greenland) during the melting season (Doherty et al., 2013). They suggested a ~~general~~ rough range of 10% to
30% in the study area. The method adopted in previous studies requires continuous sampling for about 2–3 weeks at each
site~~,~~ and thus is laborious to ~~be used~~apply for large-scale measurements in the polar area. Here, as an alternative, an
experimental approach for calculating MSC i~~w~~as proposed which may provide a new way for MSC measuring, and a further
comparison between the regional differences of MSCs is presented as well.

The melt-refreeze ice layer within the snowpack was result~~ing~~s from the refreezing of meltwater ~~that~~ percolat~~ing~~es into the
snow~~,~~. The suspended particles, especially those with larger surface areas, such as BC, may stay in place and freeze in the
crystal lattice during the refreezing of meltwater (Novotny et al., 2002). That said, the freezing process does not
preferentially exclude BC. Accordingly, here we assume that the BC concentration in the ice layer is identical to that in the
meltwater.  ~~and thus the concentration of BC in the ice layer can represent the BC values in the meltwater departing the snow.~~
The BC concentrations in the melt-refreeze ice layer and ~~in~~ its overlying snow layer were together to determine the MSC
~~associately~~ considering the thickness and density of the two layers. We conducted ~~The~~ the field measurements and sampling
~~were conducted~~ in Elson Lagoon, the Chukchi Sea and Canada Basin during the melt season (Fig. 1). After constraining the
uncertainties of this new method, the estimated MSC ~~was~~ is compared to th~~o~~se ~~values~~ derived from the RS method in the
same area, and further, the spatial variability of MSC in the western Arctic will be discussed.

**2 Field measurements and sample analysis**

We collected the snow samples in Elson Lagoon northeast of Barrow (Barrow expedition), in the Chukchi Sea (Barrow
expedition) and in the Canada Basin (1st South Korean Arctic Ocean expedition) during the late spring and summer over the
past decade (2010 to 2018). The snow physic~~cal characteristi~~cs ~~(including the snow thickness, stratification and density)~~
were also measured during the three Barrow sea ice expeditions (the year 2015, 2017 and 2018) and the 1st South Korean
Arctic Ocean expedition (the year 2010). In the 3rd Chinese Arctic expedition (the year 2008), only snow physics ~~(thickness,~~
~~stratigraphy and density)~~ were observed.

The field measurement involves the snow thickness, snow density and stratification. In Elson Lagoon, we measured the snow depth along a 10km line before melt onset (April 15, 2015), and determined the average snow depth in this region. A far-shore site was chosen ~12 km away from the coast where the snow depth was close to the mean value (31.6 ± 5.4 cm) of this region (Fig. 1). The snow stratification was firstly recorded, and then snow density was measured at 2.5 cm vertical resolution using SnowFork instrument. Four points were measured per time in each layer. We applied the average value of snow density to characterize the snow layer. The snow depth was recorded at ablation stakes next to the snow pit. In the Chukchi Sea, the spatial variation of snow depth is more significant as compared with the Elson Lagoon due to the presence of ice ridge. We firstly selected a relatively smooth area of sea ice, and measured the snow depth along a 200m line in the centre region of the flat ice on April 6th, 2017. The observation site was chosen at a location close to the average snow depth, and the measurement procedure was the same as that applied in Elson Lagoon. Note that there was a deviation between the observation sites of 2017 and 2018 due to the interannual variation in the ice condition over the Chukchi Sea (Fig .1). In Canada Basin, we conducted the measurements of snow depth at a 100m line over floe ice due to the smaller ice size and limited operating time. Snow density was measured using Tel-Tru densitometer (Tel-Tru Manufacturing Co., Inc., Rochester, NY) with an accuracy of 1 g, and a snow shovel of 2.5-cm in thickness. The thickness of the snow layer and the position of melt-refreeze ice layer were measured using a ruler.

The sample collection was performed at three stages in Elson Lagoon and the Chukchi Sea during the expeditions in 2015 and 2017. At the stage before snow-melting onset, we collected snow from 4 cm above the sea ice up to the snow surface. At the early stage of melting, the upper snow layer was firstly collected, and then the underlying ice layer was sampled separately in the same snow pit. The newly fallen snow was also collected during theonce new snowfall occurred. In order to study the spatial distribution of BC, we dug up three snow pits to sample parallelly at each site (50 meters apart from each other) and measured the physical characteristics synchronously. Observations show that the differences in BC concentrations of the three snow pits are negligible, as the standard deviation value was one order of magnitude lower than the mean concentration. We took the average BC concentration from all three pits as the BC concentration at that exact site. In At the end of the snow-melting season and when most of the snowpack had melted, we collected the top 4-cm layer of snow to analyze the BC concentration in the melted snow. In 2018, we just collected samples of melting snow in the Chukchi Sea. Table 1 shows Morethe more details of sample collecting are showed in Table S1.

Sampling was performed using a pre-cleaned plastic shovel and single-used vinyl gloves. Samples were stored in polyethylene bags that had been thoroughly washed with abundant deionized ultrapure water in the laboratory prior tobefore use. In the laboratory, the snow samples were allowed to melt in ambient temperature (18–20 ºC) and immediately filtered through quartz-fibre filters (25 mm, Whatman® QM-A). The filters were stored in an insulated cabinet with blue ice and kept in low temperature, avoiding any bacteria to produce and transited to the laboratory in the University of Chinese

| Academy of Sciences for analysis.

We used two analytical methods to measure the concentration of BC. The quartz filters were firstly dried between 60 ºC and 70 ºC and then measured using an optical transmission analytical method (Model OT-21, Magee Scientific, California USA). The OT-21 is widely used in the measurement of atmospheric BC aerosol. A~~Thereaf~~ter that, a 1.0 cm$^2$ punch was cut from

105 | each filter~~,~~ and was analyzed for elemental carbon (EC) using the "Thermos-optical NIOSH 5040" method (Sunset Laboratory Inc., Forest Grove, U.S., which has been applied to measure EC in Svalbard snow (Forsström et al., 2013). A comparison between EC and BC in a previous study (Dou et al., 2017) showed that the values obtained from two different methods are highly correlated ($R^2$ = 0.97). For consistency, we adopt BC referring to BC and EC. Five blank ~~filter~~s were processed following the same analytical procedure as the samples, except that they were filtered with ultrapure water. The

110 | measured BC background of the filters (0.03±0.02 ng g$^{-1}$) ~~are~~ is an order of magnitude lower than the concentration of the ice layer. The values in Table S1 ~~and Table S2~~ have been corrected by excluding blank ~~contributions~~.

### 3 Results and discussion

During two Arctic Ocean Expeditions (the year 2008 and 2010), ice layers developed in almost all snowpacks over sea ice in the measurement area, and the snow stratigraphy and thickness exhibited highly spatial variabilities. The observed thickness

| of ice layers ranges from ~0.3 cm to ~2.8 cm. During the field measurements in Elson Lagoon in 2015, we recorded that the ice layer came into being on May 18[th] and May 22[th], the early stage of the sea-ice melting season. The ice layer was observed in the Chukchi Sea on May 25[th]–28[th], 2017, and on May 30[th] – 31[th], 2018.

The ice layer results from the refr~~eeo~~zen meltwater that percolates into cold snow along with layer-parallel capillary barriers

| by heat conduction into surrounding subfreezing snow (Pfeffer et al., 1998; Massom et al., 2001; Colbeck et al., 2009). It detains BC particles in the meltwater, leaving the upper snow layer. Except for the formation mechanism mentioned above, ice layers could also generate from the radiation crust or liquid precipitation re-freezing (Massom et al. 2001). However, the BC concentrations in these two types of ice layers are in the same order of magnitude as those of new or recently-fallen snow. Besides, the radiation crust usually forms on the snow surface (Colbeck et al., 2009; Dou et al., 2013). The ice layer frozen

| from liquid precipitation is mostly formed during winter season before the snow-melt onset (Sturm et al., 2002; Langlois et al., 2017). These two types of ice layers cannot reflect the BC scavenging with meltwater~~,~~ and thus were not considered in this study.

By measuring BC in the selected melt-refreeze ice layer and its overlying snow, ~~it can be drawn~~we observed that the

130 | concentration of the ice layer is 0.42±0.08 ng g$^{-1}$ in the measurement area, ~~meaning~~ suggesting that ~0.42 ng of BC particles can be carried away from the snow layer by 1 gram water. Before estimating MSC, we compared the BC concentration ~~of~~ in

the ice layer ~~is also compared~~ with those of other snow layers in the measurement area at different ablation stages. The BC concentration increased from $1.32\pm0.20$ ng g$^{-1}$ in the new snow to $2.42\pm0.63$ ng g$^{-1}$ in the generally melting snow (Fig. S~~1~~2), and the concentration in the surface layer increased up to $15.91\pm1.12$ ng g$^{-1}$ ~~in~~ at the end of snow ablation.

The MSC is estimated based on the observations of BC, snow density and thickness. By determining the burden of BC per area (ng BC/cm$^2$) in the ice layer and the average original BC mass per unit area in the unmelted snowpack, the scavenging efficiency (MSC) is given by

~~According to the definition by Flanner et al. (2007), MSC can be given by:~~

$$MSC = h_i \cdot \rho_i \cdot C_{bi} / h_1 \cdot \rho_1 \cdot C_{b1} \qquad (1)$$

where $h_i$ (cm), $\rho_i$ (g cm$^{-3}$) and $C_{bi}$ (ng g$^{-1}$) are respectively the thickness, density and BC mass concentration of the ice layer (Fig. S~~2~~3); $h_1$ (cm), $\rho_1$ (g cm$^{-3}$) and $C_{b1}$ (ng g$^{-1}$) are the same variables but for the snow layer before the melt event (Fig. S~~2~~3). Note that determining scavenging efficiency with this method requires measuring the above factors at a given site at least twice, before and after the melt event.

If ~~the~~ snow physics and BC concentration were not measured before the melt event, we would choose another method to calculate MSC. We assumed that as the surface snow melts, BC particles scavenged by meltwater are refrozen in the melt-refreeze ice layer, that is, $h_1 \cdot \rho_1 \cdot C_{b1} = h_i \cdot \rho_i \cdot C_{bi} + h_2 \cdot \rho_2 \cdot C_{b2}$, where $h_2$ (cm), $\rho_2$ (g cm$^{-3}$) and $C_{b2}$ (ng g$^{-1}$) are respectively the thickness, snow density and BC mass concentration of the melting snow overlying the ice layer (Fig. S~~2~~3).

~~By determining the loading of BC per unit area (ng cm$^{-2}$) in the ice layer and in the partially melted snow layer above it, the scavenging efficiency (MSC) is given by:~~So that

$$MSC = h_i \cdot \rho_i \cdot C_{bi} / (h_i \cdot \rho_i \cdot C_{bi} + h_2 \cdot \rho_2 \cdot C_{b2}) \qquad (2)$$

~~In fact, t~~The assumption behind the proposed new method also implies that all of the melt-water generated from the original

snow column is conserved in the ice layer and its overlying snow. Thus, $h_1 \cdot \rho_1$ is also equal to $(h_i \cdot \rho_i + h_2 \cdot \rho_2)$ in the assumption.

Since the new method largely depends on the conservation of snow mass and BC content before and after the ablation event, we validate the above presumption using the observations that involve snow sampling both before and after the melt event at

160 ~~6~~six sites during the Barrow expeditions (Table S~~2~~1). The average of the snow density and BC concentration of the whole layer of snow were used to represent the situation $(\rho_1, C_{b1})$ of the upper part $(h_1)$ of the snow layer before ablation. Here, deviations from 100% conserved is used to measure the conservation of BC $((h_i \cdot \rho_i \cdot C_{bi} + h_2 \cdot \rho_2 \cdot C_{b2}) / h_1 \cdot \rho_1 \cdot C_{b1} - 100\%)$ and snow $((h_i \cdot \rho_i + h_2 \cdot \rho_2) / h_1 \cdot \rho_1 - 100\%)$, and to evaluate the uncertainty in the derived scavenging efficiencies. The loss of snow mass and BC content after the ablation event are both smaller than 7.0% (Fig. ~~2a~~4a), indicating that most of the

165 meltwater and BC within it was re-frozen in the ice layer and the BC content was substantially conserved. The assumption of

the proposed new method is valid in the measurement area during the sampling period.

According to Eq. (2), we estimated the MSC (MSC_2) in the measurement area and compared it with the MSC_1 calculated based on Eq. (1). The result indicates that there is a slight difference in the MSCs calculated separately by the two methods. The bias of MSC ((MSC_2-MSC_1)/MSC_1) caused by the deviation of snow and BC from 100% conserved before and after melt is small than 7.2% (Fig. ~~2b~~4b). Further analysis showed that there is no ~~obvious~~ apparent correlation between the estimated bias of MSC and the degree of snow melting (Fig. ~~2b~~4b).

With the new method, we calculated the MSC in Elson Lagoon and compared it with that estimated according to equation (2) in Doherty et al. (2013) ~~estimated by Doherty et al. (2013) in the same area~~. Results indicate that the MSC (14.5%) calculated by the new method is smaller than that (20.4%) by the method of Doherty et al. (2013) based on the observations in this study. The difference in MSCs estimated by these two methods is reasonable since the latter represents the upper limits of MSC. Our estimation is close to the average value (16.2%) derived by repeated sampling (RS) introduced by Doherty et al. (2013) in the same area and is still within its best estimation [14.0%-20.0%]. ~~The result indicates that the MSC in Elson Lagoon is 14.5%±2.6%, close to the average estimation (16.2%±8.5%) by repeated sampling (RS) introduced by Doherty et al. (2013) and is still within its best estimation [14.0%-20.0%]. Our estimation of the MSC is also broadly consistent with that adopted by Flanner et al. (2007) in their model study. They assumed that the MSC is 3% for the hydrophobic BC and 20% for hydrophilic BC, given that the total BC is a combination of the two types of BC.~~

The scavenging efficiency of BC is mainly determined by the particle size and the hydrophobicity, which is interfered with other impurities since BC usually occurs in the particles as an "internal mixture" in the Arctic (Doherty et al., 2013). These influencing factors show significant regional differences due to various sources of BC and distinguishing deposition and transport processes (Korhonen et al., 2008; AMAP, 2011; Sharma et al., 2013; Schulz et al., 2019), leading to spatial variations in MSCs, which has been confirmed by the observations at Barrow and dye-2 station, Greenland (Doherty et al., 2013). Conway et al. (1996) found that the hydrophilic BC is much more efficiently scavenged by~~with~~ meltwater than the hydrophobic one. Flanner et al. (2007) further estimated that the MSC for hydrophilic BC is about ten~~10~~ times that for hydrophobic one, meaning that the ~~variations in~~ relative ratios~~distributionfraction~~ of the two types of BC in the snow ~~with location~~ also have ~~important~~ impacts on the spatial distribution of~~difference in~~ the MSCs. ~~The MSC exhibits significant spatial variability due to the different particle sizes and hydrophilicity (Flanner et al., 2007).~~ From the observations in this study (Chukchi Sea, Elson Lagoon and Canada Basin) and the results of Doherty et al. (2013) (Elson Lagoon and Dye-2, Greenland), we investigated the spatial differences of MSC in the western Arctic. The average of the MSCs in the Canada Basin (23.6%±2.1%) is basically the same as that at the Dye-2 site, Greenland (23.0%±12.5%), while is ~~larger~~ more significant than that of Chukchi Sea (17.9%±5.0%); and Elson Lagoon has the lowest MSC (14.5%±2.6%) (Fig. ~~3~~5). We further analyzed the statistical significance of the differences in MSC at various locations. The Jonckheere-Terpstra test

indicated that it is highly significant ($p < 0.01$) for Elson Lagoon < Chukchi Sea < Canada Basin, and the Mann-Whitney U test demonstrated that the difference from each other is moderately significant ($p < 0.1$). The average of the MSCs in the western Arctic is 18.0%±3.8%.

This study proposes a new method for large-scale measurements of MSC over the Arctic sea ice. The estimation of MSC requires the existence of a melt-refreeze ice layer. However, the limited data from our measurements cannot support a more extensive investigation. We reviewed the snow stratigraphy records obtained during the 3[rd] Chinese Arctic expedition in summer 2008 and the expedition hiking through the North Pole from 88 ºN to 90 ºN in late spring 1995 (Xiao et al., 1997). The records show that the melt-refreeze ice layers were widely developing over high latitudes of the Arctic, which is also confirmed by the observations in Svalbard in late spring 2007–2009 (Eckerstorfer et al., 2011). The widely distributed melt-refreeze ice layer in the Arctic suggests broader applicability for this new method in estimating the MSC of BC in the Arctic, for example, along the cruise lines where it is not pragmatic to carry out long-term continuously sampling. Nevertheless, we need to note that a melt-season ice layer may not form in regions of intense melt, where we cannot obtain the MSC value using the proposed approach in this study.

This technique assumes that BC particles are not preferentially removed during meltwater freezing. W~~In fact, we~~ do not rule out that very few BC particles can still be discharged during this process. Thus, this assumption may result in an underestimation of the BC content in the melt~~-~~water, in turn leading to an underestimation of MSC. Besides, this method does not account for influxes of BC from snowfall during the melt season, which may also lead to an underestimation of MSC in the case of snowfall occurring after melt onset. The method provides an estimate of the average seasonal MSC but does not capture temporal variations ~~in~~ efficiently~~ey~~.

**3 Conclusions**

The MSC of BC ~~has~~ is ~~been found to be~~ much less than 100% in ~~few~~ previously few studies, leading to enhanced concentrations of BC in surface snow, lowering albedo and accelerating the rate of snow melting. This study proposes a new experimental approach to determine the MSC by sampling the melt-refreeze ice layer and its overlying snow in the snowpits during the melting season, assuming the complete conservation of snow and BC content before and after the ablation event. The method is different from the established methods which require repeated sampling (RS method) over an extended period. The present observations confirm that the theory adopted in the proposed method is valid in the study area, and the estimation bias of the calculated MSCs is not dependent on the melting degree during the ablation.

Further estimation with the new method demonstrated that the MSC exhibits regional differences in the western Arctic. In the measurement period, the average MSC in Canada Basin is the largest, which is close to that estimated in Greenland,

followed by those in the Chukchi Sea and ~~in~~ Elson Lagoon. The spatial difference is suggested to be considered in the future simulation of BC-in-snow over the sea ice, rather than setting MSC as a constant in the snow and sea ice model. Combined with all available observations, we estimated an average of ~~-~~MSC in the western Arctic of 18.0%±3.8% ranging from 13.0% to 30.0%.

*Data availability.* The observations ~~of snow thickness, snow density and BC concentrations applied in this study are available as the Supplement.~~ are shown in Table 1.

*Author contribution.* TD designed the experiments and performed the analyses. TD, ZD, SL, YZ, QZ, MH and CL conceived field measurements and snow sampling. All authors participated in the writing of the paper.

*Competing interests.* The authors declare that they have no conflict of interest.

*Acknowledgments.* This study is funded by the National Key Research and Development Program of China (2018YFC1406103), the National Nature Science Foundation of China (NSFC Grants~~,~~ No. 41971084 and No.~~,~~ 41425003~~, 41401079~~) and the Key Project of CAMS (KJZD-EW-G03-04). We appreciate the State Key Laboratory of Cryosphere Science of the Chinese Academy of Sciences to supply the accommodation and ice logistics support during the visit in Barrow. We also thank UIC Corporation for providing the logistic support for the field measurements over sea ice.

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

Table 1: ~~The BC~~ **The** ~~BC~~ concentrations **of BC** observed in the melt-refreeze ice layer and its overlying snow. The thickness of snow and ice layer, and snow density observed simultaneously are shown. Note that the observations before and after the ablation events in Elson Lagoon and **the** Chukchi Sea during **the** Barrow expedition are shown as site1-site6. The sampling locations and dates are also shown. Refer to figure 3 for the description of variable names. 'BC (surface melting snow)' denotes BC concentration in the top 4-cm layer of melting snow **at** ~~in~~ the end of melt season when most of the snowpack had melted.

| Sampling area | Site | Lat (°N) | Lon (°W) | $h_1$ (cm) | $\rho_1$ (g/cm³) | $C_{b1}$ (ng/g) | Sampling date | $h_i$ (cm) | $C_{bi}$ (ng/g) | $h_2$ (cm) | $\rho_2$ (g/cm³) | $C_{b2}$ (ng/g) | Sampling date | BC(Surface melting snow, ng/g) | Sampling date | Expedition |
|---|---|---|---|---|---|---|---|---|---|---|---|---|---|---|---|---|
| Elson Lagoon | 1 | 71.32 | 156.37 | 5.5 | 0.32 | 1.72 | April 26, 2015 | 0.7 | 0.36 | 3.0 | 0.36 | 1.72 | May 18, 2015 | 14.9 | May 31, 2015 | Barrow Expedition |
| Elson Lagoon | 2 | 71.32 | 156.37 | 5.4 | 0.30 | 1.70 | April 30, 2015 | 0.8 | 0.31 | 2.5 | 0.35 | 1.70 | May 22, 2015 | 15.3 | May 31, 2015 | Barrow Expedition |
| Elson Lagoon | 3 | 71.32 | 156.38 | 10.9 | 0.32 | 1.11 | May 7, 2015 | 1.7 | 0.41 | 5.0 | 0.35 | 1.98 | May 22, 2015 | 17.9 | May 31, 2015 | Barrow Expedition |
| Chukchi Sea | 4 | 71.37 | 156.54 | 11.3 | 0.31 | 2.11 | April 15, 2017 | 1.8 | 0.48 | 5.0 | 0.36 | 2.11 | May 25, 2017 | 16.1 | June 5, 2017 | Barrow Expedition |
| Chukchi Sea | 5 | 71.37 | 156.54 | 13.2 | 0.29 | 1.82 | April 16, 2017 | 2.5 | 0.34 | 4.0 | 0.35 | 1.82 | May 26, 2017 | 16.1 | June 5, 2017 | Barrow Expedition |
| Chukchi Sea | 6 | 71.37 | 156.54 | 8.5 | 0.25 | 2.91 | May 1, 2017 | 1.0 | 0.55 | 3.0 | 0.36 | 2.91 | May 28, 2017 | 17 | June 5, 2017 | Barrow Expedition |
| Chukchi Sea | 7 | 71.37 | 156.55 | --- | --- | --- | --- | 1.5 | 0.5 | 3.0 | 0.32 | 2.43 | May 30, 2018 | 14.2 | June 10, 2018 | Barrow Expedition |
| Chukchi Sea | 8 | 71.37 | 156.55 | --- | --- | --- | --- | 0.9 | 0.36 | 2.5 | 0.29 | 2.11 | May 30, 2018 | 15.9 | June 10, 2018 | Barrow Expedition |
| Chukchi Sea | 9 | 71.37 | 156.55 | --- | --- | --- | --- | 0.5 | 0.41 | 2.0 | 0.24 | 2.33 | May 30, 2018 | 14.8 | June 10, 2018 | Barrow Expedition |
| Chukchi Sea | 10 | 71.37 | 156.55 | --- | --- | --- | --- | 1.2 | 0.43 | 3.5 | 0.31 | 2.52 | May 31, 2018 | 17.3 | June 10, 2018 | Barrow Expedition |
| Chukchi Sea | 11 | 71.37 | 156.55 | --- | --- | --- | --- | 0.4 | 0.31 | 1.0 | 0.32 | 2.14 | May 31, 2018 | 17.5 | June 10, 2018 | Barrow Expedition |
| Canada Basin | 12 | 75.03 | 159.48 | --- | --- | --- | --- | 2.8 | 0.39 | 3.5 | 0.28 | 2.93 | July 22, 2010 | --- | --- | 1[st] South Korean Arctic Expedition |
| Canada Basin | 13 | 77.98 | 159.64 | --- | --- | --- | --- | 1.7 | 0.54 | 2.5 | 0.31 | 3.81 | July 26, 2010 | --- | --- | 1[st] South Korean Arctic Expedition |
| Canada Basin | 14 | 79.51 | 160.02 | --- | --- | --- | --- | 1.9 | 0.45 | 2.5 | 0.32 | 3.32 | August 1, 2010 | --- | --- | 1[st] South Korean Arctic Expedition |

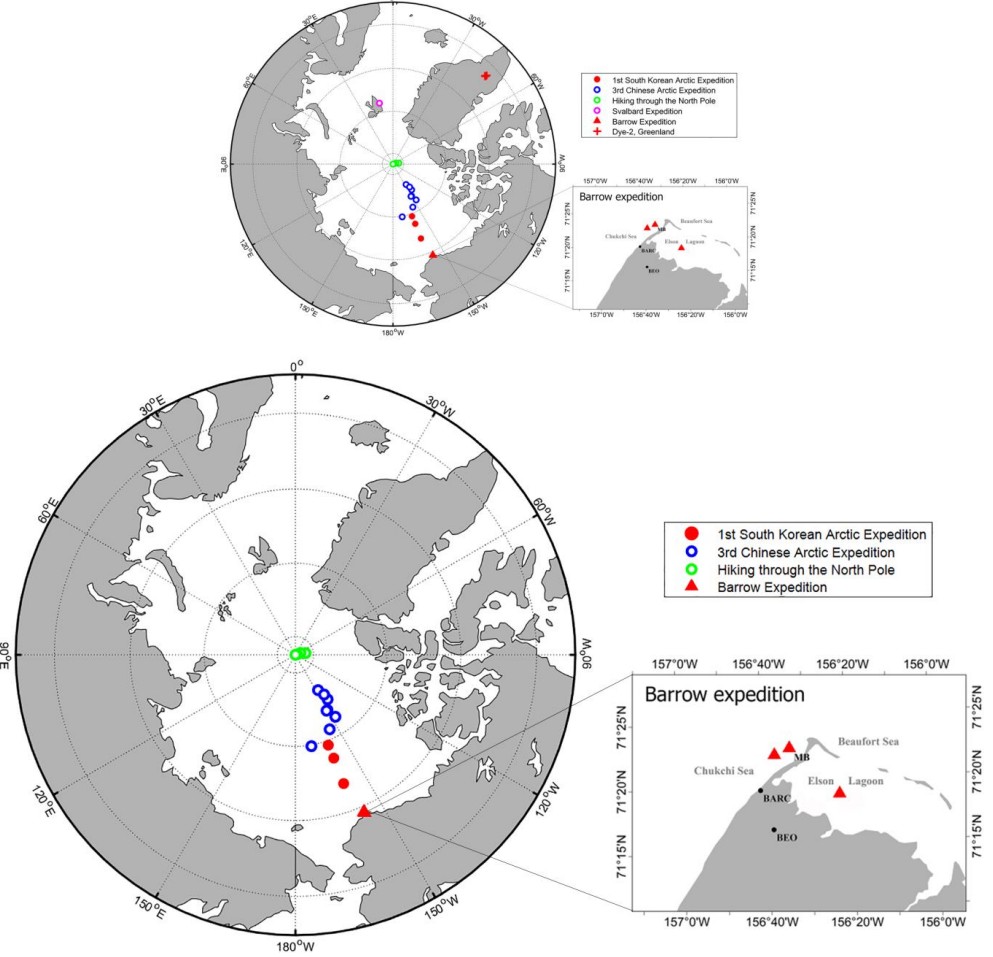

**Figure 1:** The lLocations of snow and ice layer sampling and the measurements of snow thickness and density in this study. Barrow Expeditions include the field measurements carried out in the Elson Lagoon in 2015, and in the Chukchi Sea in 2017 and 2018; the 3rd Chinese Arctic Expedition was conducted over the Canada Basin and the ~~center~~ centre region of Arctic Ocean in 2008; the 1st South Korean Arctic Expedition was conducted over the Canada Basin in 2010; the North Pole Expedition refers to the first Chinese expedition hiking through the North Pole from 88 ºN to 90 ºN in 1995 (Xiao et al., 1997)~~; the Svalbard Expedition was conducted by Eckerstorfer et al. (2011) in the field observations in 2007-2009~~. The open circle indicates the point at which the ice layer ~~wa~~is observed. ~~Solid~~ The solid triangles and ~~squares~~ circles mark the locations for both sampling and on-site measurements. ~~Cross marks the location of Dye-2 where the MSC was estimated by Doherty et al. (2013).~~

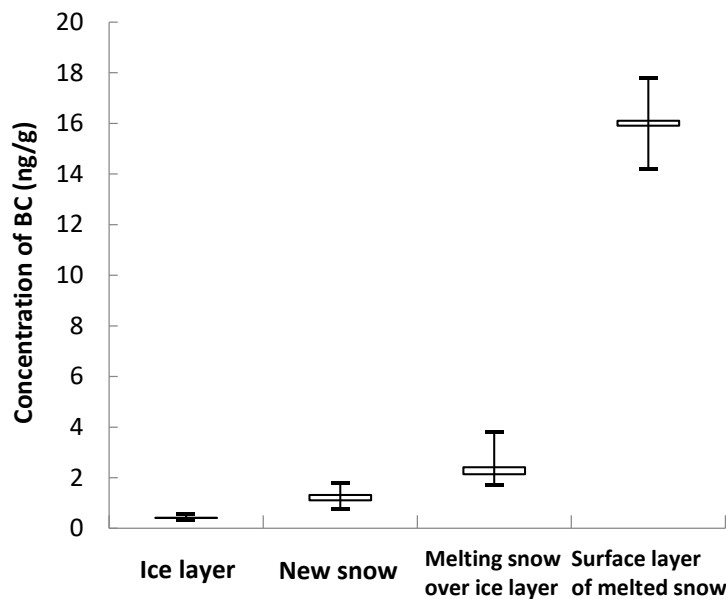

Figure 2: The BC concentrations in the melt-refreeze ice layer and melting snow, and its concentrations in the new snow and the surface layer of melting snow are also shown as a comparison. New-snow samples were only collected in Elson Lagoon and the Chukchi Sea during the measurement period. The box indicates the mean (upper) and median (bottom) values of the observations, and the whiskers constrain the full extent of the observations.

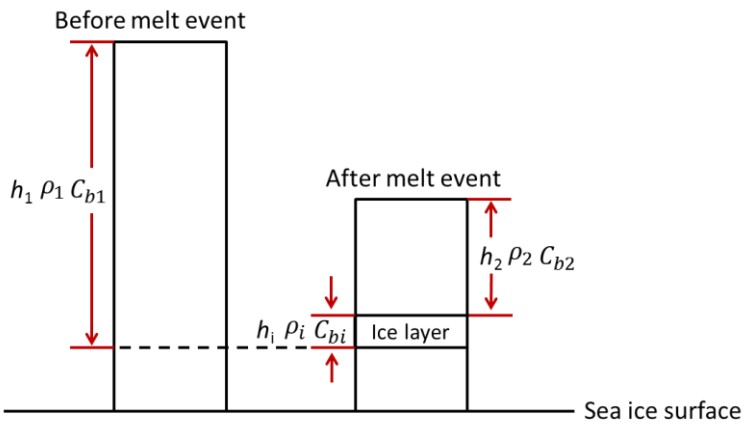

**Figure 3: Conceptual sketch of snow overlying sea ice before and after the melt event. Variables relating to the snow and ice layer mentioned in Eq. (1) and Eq. (2) are shown.**

340

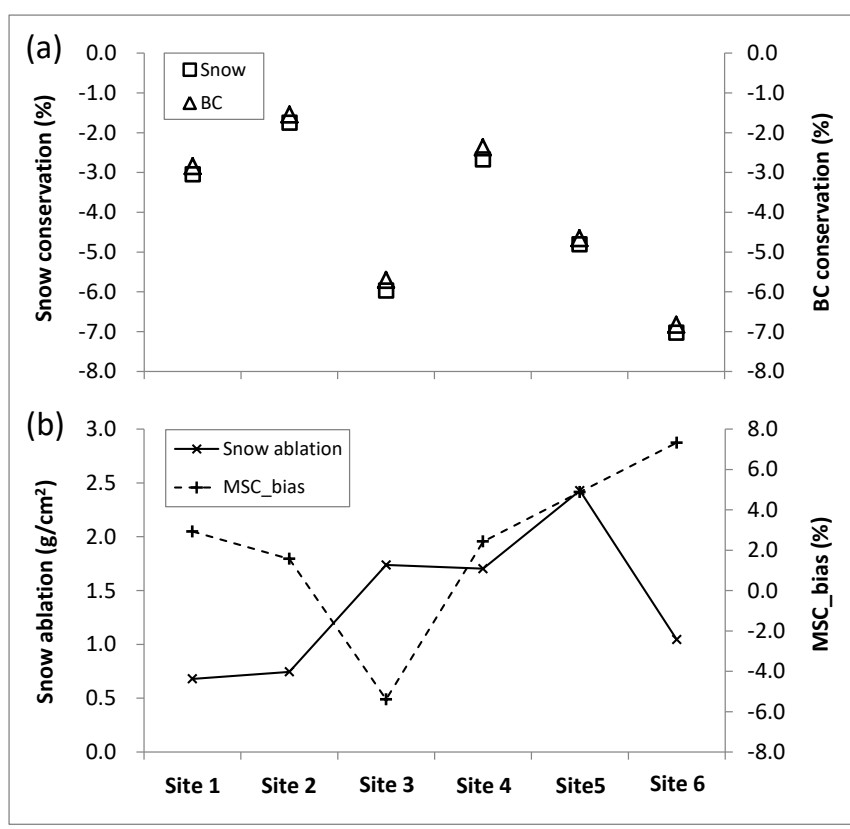

**Figure 24:** The dDeviations from 100% conserved for snow and BC after ablation (a), snow ablation ($h_1 \cdot \rho_1 - h_2 \cdot \rho_2$) during the melt event and the bias ((MSC_2-MSC_1)/MSC_1)*100% of estimated MSC based on Eq. (2) (b). The ticks on the X-axis are matching sites given in Table S21.

345

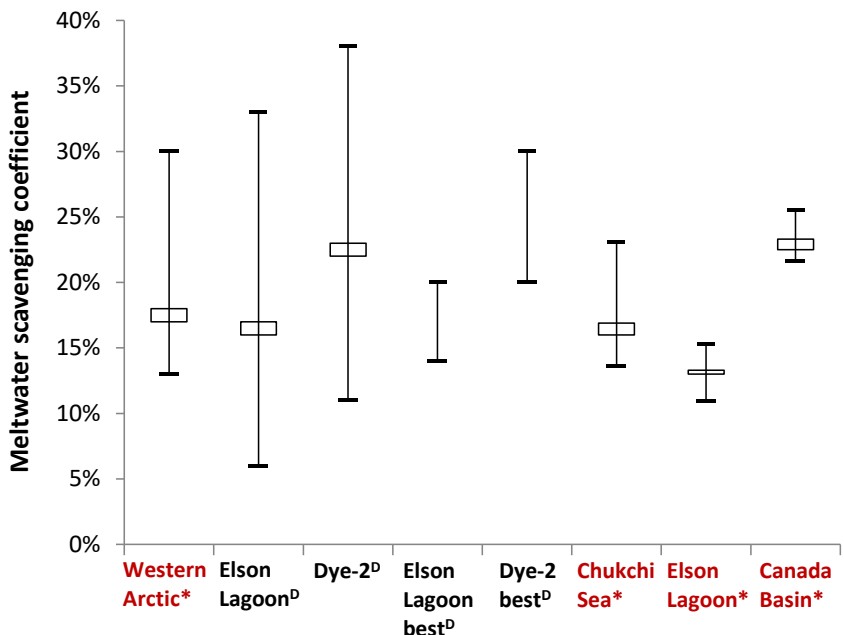

**Figure 35:** MSC of BC in different regions over the western Arctic. Superscript '*' indicates the results of this study (red), and 'D' indicates the results of Doherty et al. (2013). Elson Lagoon best[D] and Dye-2 best[D] indicate the ~~best~~ best-estimated range of MSC, respectively in Elson Lagoon and Dye-2, Greenland published in Doherty et al. (2013). The values of the western Arctic were estimated based on the observations in all measurement regions, and the ~~best~~ best-estimated values in Dye-2 and Elson Lagoon were employed in the estimation. The box ~~indicates~~ shows the mean (upper) and median (bottom) values, and the whiskers depict the extent.