# Peer review of "Brief communication: An alternative method for estimating the scavenging efficiency of black carbon by meltwater over sea ice"

_The Cryosphere, 2019_

## Referee Comment (RC1) · Anonymous Referee #1 · 24 Jul 2019

This study presents a new and efficient technique to determine the meltwater scavenging efficiency of black carbon in snow overlying sea-ice. Simply, the concentrations of BC within a melt-refreeze layer and within the overlying snow are compared, and the assumption is adopted that BC and ice have been conserved within these two layers during the melt event. Conservation within about 7% is indeed shown for a limited number of test cases, in comparison with mass measurements that were made before melt commencement. The technique is also shown to produce consistent results with those from the more rigorous repeated sampling technique employed by Doherty et al (2013). Overall, the technique described here shows promise and is attractive because of its simplicity. Several issues should be addressed prior to publication, however.

[Figure]

General issues:

- It is a bit frustrating to have to refer to supplementary figures and tables for a 'Brief Communication'. This may indicate that the material should instead be presented in a standard paper rather than a brief communication. Alternatively, could the supplemental figures (especially S2, which I think is important for conceptual understanding of the technique) be worked into the main body of the paper?

- The abstract (and manuscript, generally) should acknowledge more clearly that the technique requires the presence of a 'melt-refreeze ice layer', or some term that is similarly precise. Although the abstract refers to sampling of an "ice layer within the snowpack", it is not clear until later what the nature of this ice layer is, and confusion arises especially because the previous sentence refers to sea ice. Furthermore, a re-freeze layer will not always exist, for example when persistently warm conditions cause complete meltout of thin snow layers, and this limitation could perhaps be acknowledged more clearly.

- The applied technique also assumes that the refreezing process does not preferentially exclude BC, i.e., that the BC concentration in the ice layer will be identical to that in the melt water. Please comment on this assumption, how it could affect the utility of the technique, and any observational evidence you have that can shed light on this matter.

- Abstract, line 20: "It is concluded that MSC exhibited a regional difference in the western Arctic during the sampling period" - These differences are not very large, however (i.e., they are all substantially less than 100%, indicating inefficient scavenging of BC). Are they even statistically different from each other? If so, can you speculate on why they varied? Are there regional differences in the environment that would be expected to translate into systemic differences in the MSC? Or do these differences perhaps reflect random variability in BC properties? I suggest either downplaying these differences (because they do not appear substantial) or briefly speculating on potential

sources of such differences.

- The grammar and writing in general should be proofed by a native English speaker prior to publication.

Technical comments:

Abstract, line 15: Please specify that Elson Lagoon is near Barrow Alaska. Most readers will be unaware where the lagoon is.

Abstract, line 16: "The bias of estimated MSC ranges from ..." - What is this bias with respect to? Please briefly clarify how the 'truth' was determined.

line 58: Where was the 3rd Chinese Arctic expedition conducted? Please mention it here and also how/why these measurements are included in the present study, since they do not include BC. (i.e., the relevance of this campaign is not clear at this point in the text).

line 87: "excluding blank contributions" -> maybe "subtracting blank contributions" ?

line 98-99: "However, the BC concentrations in these two types of ice layers are in the same order of magnitude as those of new or recently-fallen snow." - But meaning what? That these types of layers could be identified via their measured BC concentration? This needs better context.

Similarly, line 102: "These two types of ice layers ... were not considered in this study." What criteria were used to identify and discard these layers? The paragraph starts out well, by describing conditions in general that can lead to ice layers, only some of which are useful for measuring MSC. Please elaborate.

line 106: Where and when was this BC concentration measured?

line 141: What do the subscripts for MSC mean? And why is MSC_2 included in parenthesis? Please clarify this notation.

lines 148-153: Do the +/- values represent variability or uncertainty? I assume the former, but please specify.

line 187: missing URL

Figure 3: Please denote the technique that was used to determine these values, i.e., repeated sampling or the ice layer approach?

---

## Referee Comment (RC2) · Howard Conway (Referee) · 31 Jul 2019

Others have observed and discussed the seasonal influence of black carbon in melting snow (e.g. Flanner et al, 2007; Doherty et al., 2010, 2013; Dou et al., 2017). Here the authors propose a new method to estimate meltwater scavenging of black carbon using field measurements made near Barrow, Alaska. The idea is to eliminate the need for repeat sampling at sites by collecting just one profile during the melt season, and assuming mass conservation to estimate the scavenging efficiency.

Some comments and questions:
1. Figure 1 shows locations of sites far from Barrow that are not used to support your new technique for estimating BC scavenging efficiency. I do think that referencing these extra sites: South Korean Antarctic Ocean expeditions 2010 -18; 3$^{rd}$ Chinese Arctic Expedition; Hiking through the North Pole; Dye 2 etc) adds to the primary focus of your paper (i.e. meltwater scavenging of BC). I think it would be less confusing for readers if these references were not included.

2. Not all readers will be familiar with your field measurements near Barrow, Alaska (Dou et al., 2017). It would be useful to expand on details of those measurements here. Also in the abstract it would be important to state specifically that Elson Lagoon and Chukchi Sea are in the vicinity of Barrow, Alaska.

3. I do not follow your eqn. 1, which you mention comes from Flanner et al (2007). Their eqn. 3 assumes mass rate of change of BC in layer 'i' is proportional to its mass mixing ratio multiplied by a scavenging factor, which appears to be quite different to the one you are using.

Instead, it might be better to follow the method described by eqns. 2, 3 and 4 in Doherty et al., (2013), which shows how measurements of $m_B$ (average original mass per unit volume of BC before melting), and $m'_B$ (average mass per unit volume in the near surface snow can be used to calculate an average scavenging efficiency. In your case. you could use the method you propose to calculate $m_B = h_1 \rho_1 C_{B1}$, and calculate $m'_B$ from your measurements of $C_{Bsfc}$ $h_{sfc}$ and $\rho_{sfc}$ in the near surface snow after melt has started (from your Table 2).

4. I am confused by the data presented in Tables S1 and S2 and how they have been used to construct Figs. 2, 3, & S1. Some questions:
(i) are there 3 different sites at Elson Lagoon that were sampled in 2015 only, and another 3 different sites at Chukchi Sea that were sampled in 2017 and again in 2018?

(ii) if so, are data for the first site at Elson Lake shown in Table S1: BC concentrations (ng/gm) of 0.31 in the ice layer and 1.72 in the overlying snow layer measured on May 18. Does that imply melt had started by that time? On May 31, an average value of 14.9ng/g was measured in the near surface snow. What is the density and thickness of the surface layer?

(iii) For the first site at Elson lake in Table S2: $C_{B1}$=1.1, and presumably $C_{B2}$ = 1.72; $C_{Bice}$ = 0.31 and $C_{Bsfc}$ = 14.9. What is the density and thickness of the surface layer?

When were the measurements of $h_1$, $\rho_1$ and $C_{B1}$ made?

(iv) Perhaps it would be less confusing if the tables were combined with relevant data (depths, densities, concentrations for Elson Lake and Chukchi Sea) and included as a single table in the main text.

5. It would be good to mention some of the limitations of the method – for example, (i) a melt-season ice layer may not form in regions of strong melt; (ii) the model does not account for influxes of BC from snowfall during the melt season; (iii) the model provides an estimate of the average seasonal scavenging efficiency but does not capture temporal variations in efficiency.

---

## Referee Comment (RC3) · Howard Conway (Referee) · 3 Aug 2019

Sorry, my first comment and questions should read:

....... I do NOT think that referencing these extra sites: South Korean Antarctic Ocean expeditions 2010 -18; 3rd Chinese Arctic Expedition; Hiking through the North Pole; Dye 2 etc) adds to the primary focus of your paper (i.e. meltwater scavenging of BC). ........

---

## Author Comment (AC1) · 30 Sep 2019

- Response to reviewer1's comments: We thank the reviewer for a helpful review. The reviewer's comments have guided further improvements in the logic and statement, making this work more rigorous. A detailed response follows below.

This study presents a new and efficient technique to determine the meltwater scavenging efficiency of black carbon in snow overlying sea-ice. Simply, the concentrations of BC within a melt-refreeze layer and within the overlying snow are compared, and the assumption is adopted that BC and ice have been conserved within these two layers during the melt event. Conservation within about 7

General issues: - It is a bit frustrating to have to refer to supplementary figures and tables for a 'Brief Communication'. This may indicate that the material should instead be presented in a standard paper rather than a brief communication. Alternatively, could the supplemental figures (especially S2, which I think is important for conceptual understanding of the technique) be worked into the main body of the paper?

- Response: Thank you for your suggestion. The supplemental figures and tables have been merged into the main body in the revised MS. A 'brief communication' has also been changed to a standard paper since the revised MS beyond the volume required by a "brief communication".

- The abstract (and manuscript, generally) should acknowledge more clearly that the technique requires the presence of a 'melt-refreeze ice layer', or some term that is similarly precise. Although the abstract refers to sampling of an "ice layer within the snowpack", it is not clear until later what the nature of this ice layer is, and confusion arises especially because the previous sentence refers to sea ice. Furthermore, a re-freeze layer will not always exist, for example when persistently warm conditions cause complete meltout of thin snow layers, and this limitation could perhaps be acknowledged more clearly.

- Response: We have clarified the melt-refreeze ice layer that formed by refreezing of the snow meltwater within the snow cover in the revised abstract and MS. A discussion of the limitation in this technique is included in the section of "Results and discussion", please see details in L187-206 in the revised MS.

- The applied technique also assumes that the refreezing process does not preferentially exclude BC, i.e., that the BC concentration in the ice layer will be identical to that in the melt water. Please comment on this assumption, how it could affect the utility of the technique, and any observational evidence you have that can shed light on this matter.

- Response: In theory, melt-water can release some BC impurities during freezing,

resulting in less BC mass concentration in the melt-refreeze ice layer than in the melt-water. However, we have no observational information about this process, so we cannot give a quantitative discussion about it in this study. We make a discussion about this uncertainty in the section of "Results and discussion", please see details in L200-203 in the revised MS.

- Abstract, line 20: "It is concluded that MSC exhibited a regional difference in the western Arctic during the sampling period" - These differences are not very large, however (i.e., they are all substantially less than 100

- Response: Thank you for your suggestion. We downplayed the regional differences in the revised abstract and MS, and some statements that emphasized the significance of regional differences were removed. The observed differences in MSC may be due to the different particle sizes and hydrophilicity of BC particles in different regions. We add a discussion about this point in the revised MS (See L177-179).

- The grammar and writing in general should be proofed by a native English speaker prior to publication.

- Response: The grammar and writing have been proofed.

Please also note the supplement to this comment:
https://www.the-cryosphere-discuss.net/tc-2019-147/tc-2019-147-AC1-supplement.pdf

———————————————

---

## Author Comment (AC2) · 30 Sep 2019

- Response to reviewer1's comments: We thank the reviewer for a helpful review. The reviewer's comments have guided further improvements in the logic and statement, making this work more rigorous. A detailed response follows below.

This study presents a new and efficient technique to determine the meltwater scavenging efficiency of black carbon in snow overlying sea-ice. Simply, the concentrations of BC within a melt-refreeze layer and within the overlying snow are compared, and the assumption is adopted that BC and ice have been conserved within these two layers during the melt event. Conservation within about 7

General issues: - It is a bit frustrating to have to refer to supplementary figures and tables for a 'Brief Communication'. This may indicate that the material should instead be presented in a standard paper rather than a brief communication. Alternatively, could the supplemental figures (especially S2, which I think is important for conceptual understanding of the technique) be worked into the main body of the paper?

- Response: Thank you for your suggestion. The supplemental figures and tables have been merged into the main body in the revised MS. A 'brief communication' has also been changed to a standard paper since the revised MS beyond the volume required by a "brief communication".

- The abstract (and manuscript, generally) should acknowledge more clearly that the technique requires the presence of a 'melt-refreeze ice layer', or some term that is similarly precise. Although the abstract refers to sampling of an "ice layer within the snowpack", it is not clear until later what the nature of this ice layer is, and confusion arises especially because the previous sentence refers to sea ice. Furthermore, a re-freeze layer will not always exist, for example when persistently warm conditions cause complete meltout of thin snow layers, and this limitation could perhaps be acknowledged more clearly.

- Response: We have clarified the melt-refreeze ice layer that formed by refreezing of the snow meltwater within the snow cover in the revised abstract and MS. A discussion of the limitation in this technique is included in the section of "Results and discussion", please see details in L187-206 in the revised MS.

- The applied technique also assumes that the refreezing process does not preferentially exclude BC, i.e., that the BC concentration in the ice layer will be identical to that in the melt water. Please comment on this assumption, how it could affect the utility of the technique, and any observational evidence you have that can shed light on this matter.

- Response: In theory, melt-water can release some BC impurities during freezing, resulting in less BC mass concentration in the melt-refreeze ice layer than in the melt-water. However, we have no observational information about this process, so we cannot give a quantitative discussion about it in this study. We make a discussion about this uncertainty in the section of "Results and discussion", please see details in L200-203 in the revised MS.

- Abstract, line 20: "It is concluded that MSC exhibited a regional difference in the western Arctic during the sampling period" - These differences are not very large, however (i.e., they are all substantially less than 100

- Response: Thank you for your suggestion. We downplayed the regional differences in the revised abstract and MS, and some statements that emphasized the significance of regional differences were removed. The observed differences in MSC may be due to the different particle sizes and hydrophilicity of BC particles in different regions. We add a discussion about this point in the revised MS (See L177-179).

- The grammar and writing in general should be proofed by a native English speaker prior to publication.

- Response: The grammar and writing have been proofed.

Please also note the supplement to this comment:
https://www.the-cryosphere-discuss.net/tc-2019-147/tc-2019-147-AC2-supplement.pdf

**Supplement:**

[revised manuscript text omitted]

The field measurements involve snow thickness, snow density and stratification which are conducted in Elson Lagoon, the

Chukchi Sea and Canada Basin. In Elson Lagoon, we measured the snow depth along a 10km line before melt onset (April 15, 2015), and determined the average value of snow depth. A far-shore site is chosen ~12 km away from the coast with snow depth close to the mean value (31.6 ± 5.4 cm) of this region. The site location is shown in Fig. 1. The snow stratification was firstly recorded, and then snow density was measured at 2.5 cm vertical resolution using SnowFork instrument, and four points was measured per time in each layer. We applied the average value of snow density to characterize the snow layer. The snow depth was recorded at ablation stakes next to the snow pit. In the Chukchi Sea, due to the presence of ice ridge, the spatial variation of snow depth is more significant than the Elson Lagoon. We firstly selected a relatively smooth area of sea ice, and measured the snow depth along a 200m line in the centre region of the flat ice on April 6th, 2017. The observation site was chosen at a location close to the average snow depth and the measurement procedure is the same as that applied in Elson Lagoon. Note that due to the interannual variability in the ice situation over the Chukchi Sea, there was a deviation for the observation sites in 2017 and 2018 (Fig .1). In Canada Basin, we conducted the measurements at a 100m line over floe ice to determine the average snow depth due to smaller ice size and limited operating time. Snow density was measured using a Tel-Tru densitometer (Tel-Tru Manufacturing Co., Inc., Rochester, NY) with an accuracy of 1 g, and a snow shovel of 2.5-cm in depth. The thickness of snow and the position of the melt-refreeze ice layer were measured with 
[revised manuscript text omitted]

MSC shows regional variability due to the differences in hydrophilicity and particle size of BC. BC particles are more effectively removed from the melting snow in the regions with smaller particle size and larger proportion of hydrophilic BC, corresponding to greater MSC and weaker enrichment of BC in the surface snow layer.

From the observations in this study (Chukchi Sea, Elson Lagoon and Canada Basin) and the results of Doherty et al. (2013) (Elson Lagoon and Dye-2, Greenland), we investigated the spatial differences of MSC in the western Arctic. The average of the MSCs in the Canada Basin (23.6%±2.1%)  basically the same as that at the Dye-2 site, Greenland (23.0%±12.5%), while is  more significant than that of Chukchi Sea (17.9%±5.0%); and Elson Lagoon has the lowest MSC (14.5%±2.6%) (Fig. 5). The average of the MSCs in the western Arctic is 18.0%±3.8%.

This study proposes a new method for large-scale measurements of MSC over the Arctic sea ice. We need to note that there may be uncertainty in the spatial difference of MSC obtained from the observations at limited sites due to random variability in BC properties during the measurement period. The estimation of MSC, in reality, requires the existence of a melt-refreeze ice layer. However, the limited data from our measurements cannot support a more extensive investigation. We reviewed the snow stratigraphy records obtained during the 3[rd] Chinese Arctic expedition in summer 2008 and the expedition hiking through the North Pole from 88 °N to 90 °N in late spring 1995 (Xiao et al., 1997), separately. The records show that the ice layers were widely developing over high latitudes of the Arctic Ocean, which is also confirmed by the observations in Svalbard in late spring 2007–2009 (Eckerstorfer et al., 2011). The widely distributed melt-refreeze ice layer in the Arctic suggests broader applicability for this new method in estimating the MSC of BC in the Arctic, for example, along the cruise lines where it is not pragmatic to carry out long-term continuously sampling. Nevertheless, we do not rule out that a meltseason ice layer may not form in regions of intense melt. In that case, we cannot obtain the MSC value in that region using the proposed approach.

This technique assumes that the refreezing process does not preferentially exclude BC, that said, the BC concentration in the ice layer will be identical to that in the melt water. In fact, some of BC impurities may be expelled during the freezing process of melt-water, thus, this assumption may result in an underestimation of the BC content in the melt-water, in turn leading to an underestimation of MSC. Besides, this method does not account for influxes of BC from snowfall during the melt season, which may also lead to an underestimation of MSC in the case of snowfall occurring after snow-melt onset.
Since the collected ice layer represents the event of melting water refreeze at a specific period, thus, 
[revised manuscript text omitted]

---

## Author Comment (AC3) · 1 Oct 2019

We thank the reviewer for a comprehensive and helpful review. The reviewer's comments have guided further improvement in the problem statement and data interpretation. We have also reviewed the relevant literature to further support our central hypothesis and expanded the discussion of study's results. A detailed response follows below.

Others have observed and discussed the seasonal influence of black carbon in melting snow (e.g. Flanner et al, 2007; Doherty et al., 2010, 2013; Dou et al., 2017). Here the authors propose a new method to estimate meltwater scavenging of black carbon

[Figure]

using field measurements made near Barrow, Alaska. The idea is to eliminate the need for repeat sampling at sites by collecting just one profile during the melt season, and assuming mass conservation to estimate the scavenging efficiency.

Some comments and questions: 1. Figure 1 shows locations of sites far from Barrow that are not used to support your new technique for estimating BC scavenging efficiency. I do not think that referencing these extra sites: South Korean Antarctic Ocean expeditions 2010 -18; 3rd Chinese Arctic Expedition; Hiking through the North Pole; Dye 2 etc) adds to the primary focus of your paper (i.e. meltwater scavenging of BC). I think it would be less confusing for readers if these references were not included.

- Response: In order to make the MS more focused, the sites in Svalbard and Greenland in original figure 1 were removed. The observational locations in the 3rd Chinese Arctic Expedition and the first Chinese expedition hiking through the North Pole are retained because we observed ice layer over sea ice in these regions, which effectively extended the application scope of this new method.

2. Not all readers will be familiar with your field measurements near Barrow, Alaska (Dou et al., 2017). It would be useful to expand on details of those measurements here. Also in the abstract it would be important to state specifically that Elson Lagoon and Chukchi Sea are in the vicinity of Barrow, Alaska.

- Response: Thank you for your comments. The description of the field measurement was added in the revised MS (See P3, L62-77). We clarify that Elson Lagoon and Chukchi Sea are in the vicinity of Barrow in the revised abstract. Besides, the measurement locations in Elson Lagoon and Chukchi Sea have been shown in figure 1.

3. I do not follow your eqn. 1, which you mention comes from Flanner et al (2007). Their eqn. 3 assumes mass rate of change of BC in layer 'i' is proportional to its mass mixing ratio multiplied by a scavenging factor, which appears to be quite different to the one you are using.

Instead, it might be better to follow the method described by eqns. 2, 3 and 4 in Doherty et al., (2013), which shows how measurements of mB (average original mass per unit volume of BC before melting), and m'B (average mass per unit volume in the near surface snow can be used to calculate an average scavenging efficiency. In your case, you could use the method you propose to calculate mB = h1rïĄš1CB1, and calculate m'B from your measurements of CBsfc hsfc and rïĄšsfc in the near surface snow after melt has started (from your Table 2).

- Response: Thank you for your valuable comments. Using the method of Doherty et al. (2013) (Eq.2 in that reference), we recalculated the BC melt water scavenging coefficient (MSC) in Elson Lagoon based on the observations in this study. It indicates that the average value is 0.204, slightly higher than the result calculated by the method proposed in this study (0.145). It is known that Doherty et al. (2013) made two assumptions in estimating MSC of BC, leading to their estimate should be the upper limits of MSC. Thus, the difference between MSC values obtained using these two different methods is reasonable. We re-checked the definitions of MSC in different literatures, and found that we did not make this issue clear in the manuscript (MS). Due to different focuses and uses, there are differences in the form of the calculation formula. Flanner et al. (2007) gave a method that is applicable to BC simulation in the CLM model; Doherty et al. (2013) proposed a formula that is applicable to continuous sampling method. Previous studies determined MSC by comparing the BC content in snowpack before and after ablation. This study estimated the BC taken away from the melt water directly by measuring its content in the melt-refreeze ice layer (this is a feature of this study). Different methods are essentially the same. This allows comparison to the values used in the Doherty et al. (2013) and Flanner et al. (2007). In view of this, we have retained the original method in the revised MS, but the introduction of this method has been modified in the revised MS (L132-134): "By determining the burden of BC per unit area (ng BC/cm2) in the ice layer and in the partially melted snow layer above it, the scavenging efficiency estimated using the proposed approach is given by.…." The limitations and possible uncertainties of this method have been discussed in the last part of the MS according to the suggestions of the reviewer (Please see the response to the last comment). Reference Flanner, M.G., Zender, C.S., Randerson, J.T. and Rasch, P.J., Present-day climate forcing and response fromblack carbon in snow, J. Geophys. Res., 112, D11202, doi:10.1029/2006JD008003, 2007. Doherty, S. J., Grenfell, T.C., Forsström, S., Hegg, D.L., Brandt, R.E. and Warren, S. G.: Observed vertical redistribution of black carbon and other insoluble light-absorbing particles in melting snow, J. Geophys. Res. Atmos., 118, 5553-5569, doi:10.1002/jgrd.50235, 2013.

4. I am confused by the data presented in Tables S1 and S2 and how they have been used to construct Figs. 2, 3,  S1. Some questions: (i) are there 3 different sites at Elson Lagoon that were sampled in 2015 only, and another 3 different sites at Chukchi Sea that were sampled in 2017 and again in 2018? (ii) if so, are data for the first site at Elson Lake shown in Table S1: BC concentrations (ng/g) of 0.31 in the ice layer and 1.72 in the overlying snow layer measured on May 18.  Does that imply melt had started by that time? On May 31, an average value of 14.9ng/g was measured in the near surface snow.  What is the density and thickness of the surface layer?  (iii) For the first site at Elson lake in Table S2: CB1=1.1, and presumably CB2 = 1.72; CBice = 0.31 and CBsfc = 14.9. What is the density and thickness of the surface layer? When were the measurements of h1, rïĄš1 and CB1 made?  (iv) Perhaps it would be less confusing if the tables were combined with relevant data (depths, densities, concentrations for Elson Lake and Chukchi Sea) and included as a single table in the main text.

- Response: All observations of thickness and density of snow cover and BC concentration, and thickness and concentration of the melt-refreeze ice layer before and after the ablation event in different regions and the corresponding measurement date are included as a single table (Table 1) in the revised MS. The density and thickness of the surface layer were also included in the revised MS.

5. It would be good to mention some of the limitations of the method – for example, (i) a melt-season ice layer may not form in regions of strong melt; (ii) the model does not account for influxes of BC from snowfall during the melt season; (iii) the model provides an estimate of the average seasonal scavenging efficiency but does not capture temporal variations in efficiency.

- Response: Thank you a lot for your suggestion, we have included a discussion of the limitations of the method in the section of "Result and discussion", please see details in L187-206 in the revised MS.

Please also note the supplement to this comment:
https://www.the-cryosphere-discuss.net/tc-2019-147/tc-2019-147-AC3-supplement.pdf

**Supplement:**

[revised manuscript text omitted]

The field measurements involve snow thickness, snow density and stratification which are conducted in Elson Lagoon, the

Chukchi Sea and Canada Basin. In Elson Lagoon, we measured the snow depth along a 10km line before melt onset (April 15, 2015), and determined the average value of snow depth. A far-shore site is chosen ~12 km away from the coast with snow depth close to the mean value (31.6 ± 5.4 cm) of this region. The site location is shown in Fig. 1. The snow stratification was firstly recorded, and then snow density was measured at 2.5 cm vertical resolution using SnowFork instrument, and four points was measured per time in each layer. We applied the average value of snow density to characterize the snow layer. The snow depth was recorded at ablation stakes next to the snow pit. In the Chukchi Sea, due to the presence of ice ridge, the spatial variation of snow depth is more significant than the Elson Lagoon. We firstly selected a relatively smooth area of sea ice, and measured the snow depth along a 200m line in the centre region of the flat ice on April 6th, 2017. The observation site was chosen at a location close to the average snow depth and the measurement procedure is the same as that applied in Elson Lagoon. Note that due to the interannual variability in the ice situation over the Chukchi Sea, there was a deviation for the observation sites in 2017 and 2018 (Fig .1). In Canada Basin, we conducted the measurements at a 100m line over floe ice to determine the average snow depth due to smaller ice size and limited operating time. Snow density was measured using a Tel-Tru densitometer (Tel-Tru Manufacturing Co., Inc., Rochester, NY) with an accuracy of 1 g, and a snow shovel of 2.5-cm in depth. The thickness of snow and the position of the melt-refreeze ice layer were measured with 
[revised manuscript text omitted]

MSC shows regional variability due to the differences in hydrophilicity and particle size of BC. BC particles are more effectively removed from the melting snow in the regions with smaller particle size and larger proportion of hydrophilic BC, corresponding to greater MSC and weaker enrichment of BC in the surface snow layer.

From the observations in this study (Chukchi Sea, Elson Lagoon and Canada Basin) and the results of Doherty et al. (2013) (Elson Lagoon and Dye-2, Greenland), we investigated the spatial differences of MSC in the western Arctic. The average of the MSCs in the Canada Basin (23.6%±2.1%)  basically the same as that at the Dye-2 site, Greenland (23.0%±12.5%), while is  more significant than that of Chukchi Sea (17.9%±5.0%); and Elson Lagoon has the lowest MSC (14.5%±2.6%) (Fig. 5). The average of the MSCs in the western Arctic is 18.0%±3.8%.

This study proposes a new method for large-scale measurements of MSC over the Arctic sea ice. We need to note that there may be uncertainty in the spatial difference of MSC obtained from the observations at limited sites due to random variability in BC properties during the measurement period. The estimation of MSC, in reality, requires the existence of a melt-refreeze ice layer. However, the limited data from our measurements cannot support a more extensive investigation. We reviewed the snow stratigraphy records obtained during the 3[rd] Chinese Arctic expedition in summer 2008 and the expedition hiking through the North Pole from 88 °N to 90 °N in late spring 1995 (Xiao et al., 1997), separately. The records show that the ice layers were widely developing over high latitudes of the Arctic Ocean, which is also confirmed by the observations in Svalbard in late spring 2007–2009 (Eckerstorfer et al., 2011). The widely distributed melt-refreeze ice layer in the Arctic suggests broader applicability for this new method in estimating the MSC of BC in the Arctic, for example, along the cruise lines where it is not pragmatic to carry out long-term continuously sampling. Nevertheless, we do not rule out that a meltseason ice layer may not form in regions of intense melt. In that case, we cannot obtain the MSC value in that region using the proposed approach.

This technique assumes that the refreezing process does not preferentially exclude BC, that said, the BC concentration in the ice layer will be identical to that in the melt water. In fact, some of BC impurities may be expelled during the freezing process of melt-water, thus, this assumption may result in an underestimation of the BC content in the melt-water, in turn leading to an underestimation of MSC. Besides, this method does not account for influxes of BC from snowfall during the melt season, which may also lead to an underestimation of MSC in the case of snowfall occurring after snow-melt onset.
Since the collected ice layer represents the event of melting water refreeze at a specific period, thus, 
[revised manuscript text omitted]

---

## Author Comment (AC4) · 1 Oct 2019

-Response: In order to make the MS more focused, the sites in Svalbard and Greenland in original figure 1 were removed. The observational locations in the 3rd Chinese Arctic Expedition and the first Chinese expedition hiking through the North Pole are retained because we observed ice layer over sea ice in these regions, which effectively extended the application scope of this new method. Please also see the response to your main comments.

---

## Author Response (AR1)

**Response letter**

Dear Editor,

We have studied the valuable comments from yourself and the reviewers carefully, and made further revisions in the manuscript that address your and the reviewers' concerns. Our detailed response to the reviewers' comments follows below.

**- Response to reviewer#1's comments:**

We thank the reviewer for a helpful review. The reviewer's comments have guided further improvements in the logic and statement, making this work more rigorous. A detailed response follows below.

**This study presents a new and efficient technique to determine the meltwater scavenging efficiency of black carbon in snow overlying sea-ice. Simply, the concentrations of BC within a melt-refreeze layer and within the overlying snow are compared, and the assumption is adopted that BC and ice have been conserved within these two layers during the melt event. Conservation within about 7% is indeed shown for a limited number of test cases, in comparison with mass measurements that were made before melt commencement. The technique is also shown to produce consistent results with those from the more rigorous repeated sampling technique employed by Doherty et al (2013). Overall, the technique described here shows promise and is attractive because of its simplicity. Several issues should be addressed prior to publication, however.**

**General issues:**
**- It is a bit frustrating to have to refer to supplementary figures and tables for a 'Brief Communication'. This may indicate that the material should instead be presented in a standard paper rather than a brief communication. Alternatively, could the supplemental figures (especially S2, which I think is important for conceptual understanding of the technique) be worked into the main body of the paper?**
**- Response:** Thank you for your suggestion. The figures and tables in the supplementary have been merged into the main body of the revised MS.

**- The abstract (and manuscript, generally) should acknowledge more clearly that the technique requires the presence of a 'melt-refreeze ice layer', or some term that is similarly precise. Although the abstract refers to sampling of an "ice layer within the snowpack", it is not clear until later what the nature of this ice layer is, and confusion arises especially because the previous sentence refers to sea ice. Furthermore, a refreeze layer will not always exist, for example when persistently warm conditions cause complete meltout of thin snow layers, and this limitation could perhaps be acknowledged more clearly.**
**- Response:** We have clarified that the 'ice layer' here refers to the melt-refreeze ice layer that is produced from refreezing of the meltwater within the snowpack over sea ice in the revised abstract and MS.

A discussion of the limitation in this technique is included in the section of "Results and discussion", please see details in L187-206 in the revised MS.

**- The applied technique also assumes that the refreezing process does not preferentially exclude BC, i.e., that the BC concentration in the ice layer will be identical to that in the melt water. Please comment on this assumption, how it could affect the utility of the technique, and any observational evidence you have that can shed light on this matter.**
**- Response:** Thank for your advice. We have not obtained such observations during the field measurements. In theory, melting water can discharge some of impurities as it freezes, leading to BC mass concentration in the melt-refreeze

ice layer be lower than that in the melt water. Therefore, ignoring this effect may result in an underestimation of MSC, theoretically.

However, suspended particles, especially those with larger surface areas, such as BC, may stay in place and freeze in the crystal lattice during the refreezing of melt water (Novotny et al., 2002). That said, the freezing process does not preferentially exclude BC. In addition, the impurities can be more effectively discharged during multiple freeze-thaw processes, while it is limited for BC particles to be expelled in the formation of melt-refreeze ice layer during one freeze event. In conclusion, BC released during the freezing process of melt water may have less impact on the MSC estimation. We further clarify this assumption in L47-50 and made a discussion about the uncertainty in the section of "Results and discussion", please see details in L212-214 in the revised MS.

*Reference:*
Novotny, V. and P. A. Krenkel (2002), Water Quality: Diffuse Pollution and Watershed Management, 2nd Edition, Hoboken, NJ : J. Wiley, c2003.

**- Abstract, line 20: "It is concluded that MSC exhibited a regional difference in the western Arctic during the sampling period" - These differences are not very large, however (i.e., they are all substantially less than 100%, indicating inefficient scavenging of BC). Are they even statistically different from each other? If so, can you speculate on why they varied? Are there regional differences in the environment that would be expected to translate into systemic differences in the MSC? Or do these differences perhaps reflect random variability in BC properties? I suggest either downplaying these differences (because they do not appear substantial) or briefly speculating on potential sources of such differences.**

**- Response:** Thank you for your suggestion. We analyzed the statistical significance of the differences in MSC at various locations. The Jonckheere-Terpstra test indicated that it is highly significant ($p < 0.01$) for Elson Lagoon < Chukchi Sea < Canada Basin, and the Mann-Whitney U test demonstrated that the difference from each other is moderately significant ($p < 0.1$). This analysis has been added in the revised MS (see L196-199).

The scavenging efficiency of BC is mainly determined by the particle size and the hydrophobicity, which is interfered with other impurities since BC usually occurs in the particles as an "internal mixture". These influencing factors show significant regional differences due to various sources of BC and distinguishing deposition and transport processes (AMAP, 2011; Korhonen et al., 2008; Sharma et al., 2013; Schulz et al., 2019), leading to spatial variations in MSCs, which has been confirmed by the observations at Barrow and dye-2 in Doherty et al. (2013). The sources and properties of BC aerosols at Barrow and Canadian Arctic (Alert station) are also significantly different (Sharma et al., 2006). Conway et al. (1996) found that hydrophilic BC is much more effectively scavenged with melt water than hydrophobic one. Flanner et al. (2007) further estimated that the MSC for hydrophilic BC is about 10 times that for hydrophobic one, meaning that the variations in fraction of the two in the snow with location will also have important impacts on the spatial difference in MSCs.

There is significant seasonal variation in the size distribution of BC particles in the Arctic. In contrast, few studies have been done on the spatial difference in the particle size due to lack of observations, and only a range of 160 and 220 nm has been reported in the Arctic at present (Schulz et al., 2019).

In conclusion, we speculated that the regional differences in the hydrophobicity and particle size may cause systemic differences in the MSC, which still needs further observations to confirm. We briefly discussed the potential factors those may lead to MSC spatial differences in the revised MS (See L177-179):

*"The scavenging efficiency of BC is mainly determined by the particle size and the hydrophobicity, which is interfered with other impurities since BC usually occurs in the particles as an "internal mixture" in the Arctic (Doherty et al., 2013). These influencing factors show significant regional differences due to various sources of BC and distinguishing deposition and transport processes (Korhonen et al., 2008; AMAP, 2011; Sharma et al., 2013; Schulz et al., 2019), leading to spatial variations in MSCs, which has been confirmed by the observations at Barrow and dye-2 station, Greenland*

*(Doherty et al., 2013). Conway et al. (1996) found that the hydrophilic BC is much more efficiently scavenged with meltwater than the hydrophobic one. Flanner et al. (2007) further estimated that the MSC for hydrophilic BC is about 10 times that for hydrophobic one, meaning that the variations in fraction of the two in the snow with location will also have important impacts on the spatial difference in MSC."*

- Response: The samples were collected at two different sites in Elson Lagoon in 2015. At one of the sites (71.3SN, 156.37W), we took samples at different dates. More details could be seen in Table 1 in the revised MS. At Chukchi Sea, the samples were gathered at two different sites selected in 2017 and 2018, respectively.

**(ii) if so, are data for the first site at Elson Lake shown in Table S1: BC concentrations (ng/g) of 0.31 in the ice layer and 1.72 in the overlying snow layer measured on May 18. Does that imply melt had started by that time? On May 31, an average value of 14.9ng/g was measured in the near surface snow. What is the density and thickness of the surface layer?**

- Response: Yes, snow has begun to melt on May 18, 2015. The density and thickness of the surface layer have been included in Table 1 in the revised MS. The surface melting snow in the tables refer to the surface layer of snow pack at the final stage of snow melting.

**(iii) For the first site at Elson lake in Table S2: CB1=1.1, and presumably CB2 = 1.72; CBice = 0.31 and CBsfc = 14.9. What is the density and thickness of the surface layer? When were the measurements of h1, r☒1 and CB1 made?**

- Response: The density and thickness of the surface layer are indicated by $\rho_2$ and $h_2$ in Table 1 in the revised MS, and the measurement dates are also shown aside.

**(iv) Perhaps it would be less confusing if the tables were combined with relevant data (depths, densities, concentrations for Elson Lake and Chukchi Sea) and included as a single table in the main text.**

- Response: Thank you for your suggestion. All observations of thickness and density of snow cover and BC concentration, and thickness and concentration of the melt-refreeze ice layer in different regions and the corresponding measurement date are included as a single table (Table 1) in the revised MS.

**5. It would be good to mention some of the limitations of the method – for example, (i) a melt-season ice layer may not form in regions of strong melt; (ii) the model does not account for influxes of BC from snowfall during the melt season; (iii) the model provides an estimate of the average seasonal scavenging efficiency but does not capture temporal variations in efficiency.**

- Response: Thank you a lot for your suggestion, we have included a discussion of the limitations of the method in the section of "Result and discussion", please see details in L208-217 in the revised MS.

[revised manuscript text omitted]

**Figure 1: Locations of snow and ice layer sampling and the measurements of snow thickness and density in this study. Barrow Expeditions include the field measurements carried out in the Elson Lagoon in 2015, and in the Chukchi Sea in 2017 and 2018; the 3rd Chinese Arctic Expedition was conducted over the Canada Basin and the  centre region of Arctic Ocean in 2008; the 1st South Korean Arctic Expedition was conducted over the Canada Basin in 2010; the North Pole Expedition refers to the first Chinese expedition hiking through the North Pole from 88 ºN to 90 ºN in 1995 (Xiao et al., 1997). The open circle indicates the point at which the ice layer is observed.  The solid triangles and  circles mark the locations for both sampling and on-site measurements. **

[Figure]

Figure 2: The BC concentrations in the melt-refreeze ice layer and melting snow, and its concentrations in the new snow and the surface layer of melting snow are also shown as a comparison. New-snow samples were only collected in Elson Lagoon and the Chukchi Sea during the measurement period. The box indicates the mean (upper) and median (bottom) values of the observations, and the whiskers constrain the full extent of the observations.

330

[Figure]

335 **Figure 3: Conceptual sketch of snow overlying sea ice before and after the melt event. Variables relating to the snow and ice layer mentioned in Eq. (1) and Eq. (2) are shown.**

[Figure]

**Figure 2̶4:  Deviations from 100% conserved for snow and BC after ablation (a), snow ablation ($h_1 \cdot \rho_1 - h_2 \cdot \rho_2$) during the melt event and the bias (($MSC\_2-MSC\_1)/MSC\_1)*100\%$ of estimated MSC based on Eq. (2) (b). The ticks on the X-axis are matching sites given in Table S̶2̶1.**

340

[Figure]

**Figure 3̶5:** MSC of BC in different regions over the western Arctic. Superscript '*' indicates the results of this study (red), and 'D' indicates the results of Doherty et al. (2013). Elson Lagoon best[D] and Dye-2 best[D] indicate the  best-estimated range of MSC, respectively in Elson Lagoon and Dye-2, Greenland published in Doherty et al. (2013). The values of the western Arctic were estimated based on the observations in all measurement regions, and the  best-estimated values in Dye-2 and Elson Lagoon were employed in the estimation. The box  shows the mean (upper) and median (bottom) values, and the whiskers depict the extent.